# Pan-Cancer Analysis of Human Kinome Gene Expression and Promoter DNA Methylation Identifies Dark Kinase Biomarkers in Multiple Cancers

**DOI:** 10.3390/cancers13061189

**Published:** 2021-03-10

**Authors:** Siddesh Southekal, Nitish Kumar Mishra, Chittibabu Guda

**Affiliations:** Department of Genetics, Cell Biology, and Anatomy, University of Nebraska Medical Center, Omaha, NE 68198-5805, USA; siddesh.southekal@unmc.edu

**Keywords:** pan-cancer, kinome, dark kinase, understudied kinase, CpG methylation, correlation analysis, survival analysis, promoter, TCGA

## Abstract

**Simple Summary:**

Due to their central role in many biological processes, studies on protein kinases are an area of active research. In the last few decades, several inhibitors that target kinases have been approved by U.S. Food and Drug Administration (FDA). However, a large proportion of kinases remain uncharacterized, with very little information on their functionality. Variations in genome-wide DNA methylation and gene expression pattern have been extensively studied in many cancers. However, an extensive kinome-centered pan-cancer methylation and expression analysis are still lacking. This study aims to identify prognostic and diagnostic biomarkers, focusing on uncharacterized (dark) kinases to further encourage their research as therapeutic targets.

**Abstract:**

Kinases are a group of intracellular signaling molecules that play critical roles in various biological processes. Even though kinases comprise one of the most well-known therapeutic targets, many have been understudied and therefore warrant further investigation. DNA methylation is one of the key epigenetic regulators that modulate gene expression. In this study, the human kinome’s DNA methylation and gene expression patterns were analyzed using the level-3 TCGA data for 32 cancers. Unsupervised clustering based on kinome data revealed the grouping of cancers based on their organ level and tissue type. We further observed significant differences in overall kinase methylation levels (hyper- and hypomethylation) between the tumor and adjacent normal samples from the same tissue. Methylation expression quantitative trait loci (meQTL) analysis using kinase gene expression with the corresponding methylated probes revealed a highly significant and mostly negative association (~92%) within 1.5 kb from the transcription start site (TSS). Several understudied (dark) kinases (PKMYT1, PNCK, BRSK2, ERN2, STK31, STK32A, and MAPK4) were also identified with a significant role in patient survival. This study leverages results from multi-omics data to identify potential kinase markers of prognostic and diagnostic importance and further our understanding of kinases in cancer.

## 1. Introduction

Cancer is a heterogeneous disease that has contributed to approximately 606,520 projected deaths in the United States alone in 2020 [1]. Unfortunately, it is also a disease that is extremely challenging to treat due to the heterogeneous nature of the tumors and the lack of a variety of drugs that are effective against different tumor subtypes. Oncogene activation and the inactivation of tumor suppressors, in which protein kinases play a large role, are the major drivers contributing to cancer development [2]. The activity profile of kinases has been identified to be very distinct in tissue groups, namely, healthy, immunological and hematological, solid cancers, and mixed tissues [3]. Similarly, high tissue specificity of kinase gene expression has been shown in normal tissue datasets from GTEx [4]. Therefore, it is imperative to dissect cancer-associated kinase genes in tumor subtype and identify effective therapeutic targets.

The human protein kinases (known collectively as the kinome) represent an essential and diverse family of enzymes often dysregulated in cancer. There are around 518 known kinase genes in the human genome, and 478 of these belong to the classical protein kinase family, and 40 are atypical protein kinases [5]. Approximately 10% of them belong to the pseudokinase family, which lack catalytic activity and are distributed across all kinase families [6]. Although the human kinome is encoded by only 2% of the total coding genes, these proteins phosphorylate about 30% of the cellular proteins critical for regulation of various biological processes such as proliferation, cell cycle progression, apoptosis, motility, growth, and differentiation. Kinases are essential mediators of intracellular signal-transduction pathways and mediate many critical events such as cell fate determination and cell cycle control, making the study of kinome profiles very relevant in understanding cancer initiation and progression. Besides, dysregulation of kinase activity by events such as altered expression, copy number amplification, aberrant phosphorylation, somatic mutation, chromosomal translocation, and epigenetic regulation are also frequently oncogenic or tumor-suppressive and can be critical for disease progression and metastasis. Due to an accessible binding pocket and identified dysregulation in many diseases, including cancer, kinases are one of the most explored classes of therapeutic cancer targets. However, among the FDA-approved kinase inhibitors, many drugs inhibit off-target kinases with varying potency. These unselective interactions and inhibition of other non-kinase targets emphasize the importance of better target identification [7]. In recent years, there has been a significant improvement in understanding the role of epigenetic factors in cancer. This knowledge can be used to combine epigenetic therapies and other agents in combination therapies [8]. Recent systematic studies of the kinome drug-target interaction profiles have shown drug discovery research mainly focused on tyrosine (Tyr) kinase family [9]. In contrast, a large fraction of the kinome target space remains unexplored. This also includes the pseudokinase families, which are usually overlooked despite their links to many cancers, which can be attributed to the lack of small molecule inhibitors and assays to evaluate their mechanisms of action [6,10]. As of 2019, only 8% of FDA-approved small molecule inhibitors target kinases, most (70%) of which belong to the Tyr kinase family [11]. While significant kinase research has been focused on Tyr and Ser/Thr kinases, those belonging to other families have often been understudied, with very little information available on their role in cellular processes and their use as druggable targets. Based on an initial study by the National Institutes of Health (NIH), 134 protein kinases are classified as ‘under-studied’ or ‘dark kinases’ based on Jenson PubMed and RO1/PubTator score. Most of these dark kinases belong to the ‘other’, CMGC, Ca^2+^/calmodulin-dependent protein kinase (CaMK) groups [11,12]. Data resources such as *the dark kinase knowledgebase* (DKK), supported through NIH’s *Illuminating the Druggable Genome* (IDG) program, collates both experimental and bioinformatic data to encourage the study of understudied kinases [13,14]. Among the 162 dark kinases listed in DKK, the focus proteins include Protein Kinase, Membrane Associated Tyrosine/Threonine 1 (PKMYT1), Tousled-like kinase 2 (TLK2), BR serine/threonine kinase 2 (BRSK2), cyclin-dependent kinase 12 (CDK12), and cyclin-dependent kinase 13 (CDK13) kinases. Recent structural studies have also focused on Tyr pseudokinases such as ROR1 to understand their mechanism and target them pharmacologically [15]. Other high-throughput screening methods to identify functional analogs to pseudokinase, Kinase suppressor of Ras 1 (KSR1) (involved in EGFR-Ras-MAP kinase pathway) have been recently published [16].

Methylation of CpG sites is an epigenetic process which play a crucial role in gene expression regulation. Often methylated CpG sites are observed at high frequency in specific genomic regions called CpG islands. Several genome-wide methylome analyses and correlation analyses in cancer have revealed aberrant methylation profiles [17]. Earlier methylation studies in kinases have focused on a few selected kinase genes or gene families in specific cancer types. For instance, Kuang et al. conducted a comprehensive genome-wide analysis of the Eph/ephrin receptor tyrosine kinase family and identified 15 hypermethylated genes in acute lymphoblastic leukemia [18]. Similarly, other experimental studies have focused on the expression and promoter methylation of SRC, LYN and CKB kinases in gastric and CHK2 kinase in non-small cell lung cancers, respectively [19,20]. Although a large scale bioinformatic study has been performed on the comparison of expression levels of 459 kinase genes in 5681 normal and tumor tissues [3], there are no comprehensive pan-cancer correlative studies between gene expression and methylation data of kinome using TCGA data, to-date.

This study analyzed the global human kinome expression and corresponding promoter DNA methylation profiles across 32 TCGA cancer types to study the altered kinome expression profiles and correlate with corresponding promoter methylation status. Using this information, we identified potential kinase biomarkers with clinical relevance. Based on Cox-regression analysis and log-rank test, we further highlight the role of commonly upregulated dark kinases, PKMYT1, Pregnancy Up-Regulated Nonubiquitous CaM Kinase (PNCK), BRSK2, Endoplasmic Reticulum To Nucleus Signaling 2 (ERN2), and Serine/Threonine Kinase 31 (STK31) in survival, which are potential therapeutic targets and hence worth exploring [21]. We also demonstrate the ability of kinome expression and methylation profiles to distinguish between 30 cancer types based on unsupervised clustering analysis, further emphasizing the similarity of kinome activity within the organ systems and the tissue and histological levels.

## 2. Materials and Methods

### 2.1. Data Retrieval

The initial list of 504 human kinase genes was downloaded from UniprotKB [22]. From there, we used a set of 496 kinase genes whose data are available in the harmonized TCGA data from the GDC (Genomic Data Commons). The omitted genes include GRK2, GRK3, MAP3K20, MAP3K21, PAK5, PRAG1, COQ8A, COQ8B, whose data are not present across all the datatypes. *TCGAbiolinks* [23], a Bioconductor tool, was used to download the level-3 DNA methylation (Illumina HumanMethylation450 BeadArray), FPKM-UQ gene expression (Illumina HiSeq RNASeq V2) of 496 kinases, and clinical data of corresponding tumor samples. DNA methylation data of 7245 probes in the promoter region (+/− 1.5 Kb from the TSS) of the kinome gene set were retained. The curated annotation of the kinase family was downloaded from UniProtKB. Duplicated samples were removed from further analyses. The kinome gene set also consisted of a subset of 148 dark kinases out of the 162 understudied kinases listed in DKK. The list of kinase genes included in this study are given in Appendix A. The complete list of TCGA cancer types abbreviations can be found at https://gdc.cancer.gov/resources-tcga-users/tcga-code-tables/tcga-study-abbreviations accessed on 2 February 2021.

### 2.2. Gene Expression Data

We downloaded fragments per kilobase of transcript per million mapped reads upper quartile (FPKM-UQ) data of 496 kinases. We then removed all genes with missing expression values (for at least 25% of the samples) and genes that had CPM (count per million) numbers less than one (for at least 25% of the samples). Differential gene expression (DGE) analysis was performed using the Bioconductor tool, *limma* following log2 normalisation [24]. Benjamini-Hochberg (BH) adjusted *p*-value cut-off of 0.005, and an absolute log2 fold change of 1 was used to obtain the list of differentially expressed genes.

### 2.3. Methylation Data

The intensities of the methylated and unmethylated alleles at the analyzed CpG sites were measured using the Beta-value (β), which ranges between 0 and 1. CpGs with missing β values in >25% tumor and normal samples for each cancer were excluded from further analysis. K-nearest neighbor-based Imputation method was used to fill the missing β values using the *imputeKNN* module of the R tool, *impute* [25]. Correction of type bias was done using the beta mixed-integer quantile normalization (BMIQ) [26] using the R package, *ChAMP* [27]. CpG probes mapped against X, Y, and mitochondrial chromosomes were excluded from analyses to eliminate gender bias. We also removed the CpG probes that overlapped with repeat masker and SNPs from dbSNP v151 with minor allele frequency (MAF) >1% [28] to help remove the sequence polymorphisms that can affect DNA methylation readouts in the Infinium arrays. We retained β values of 7245 kinase-specific unique CpG probes within ±1500 bp from the TSS site for differential methylation analysis in the final annotation file. We obtained differentially methylated CpG probes with FDR 0.05 and mean β value difference of at least 0.2 (Δβ ≥ 0.2).

### 2.4. t-SNE Plots

Dimensionality reduction using the t-SNE method was carried out for the expression data of 496 kinases and corresponding 7245 BMIQ normalized β values of CpG probes at the promoter region, both individually and combined. The combined final dataset consisted of 7783 samples from 30 cancer types with kinase expression and methylation data after removing the missing data. Lymphoid Neoplasm Diffuse Large B-cell Lymphoma (DLBC) was omitted in the analysis due to lack of methylation data. Testicular Germ Cell Tumors (TGCT) was omitted as it failed in BMIQ normalization as several samples doesn’t follow beta distribution. Tumor histological subtype information was downloaded from the GDC data portal. The expression data were normalized between 0 and 1 to remove bias in the analysis. t-SNE was performed using the R package, *Rtsne,* to get the data points’ coordinates, which were colored as per the cancer types. 2D and 3D t-SNE plots were generated using R packages, *ggplot2* and *plotly,* respectively, and were colored as per the TCGA tumor and tissue subtypes.

### 2.5. Correlation Analysis

Correlation analysis between DNA methylation and corresponding gene expression was performed using meQTL based on non-zero Pearson correlation for samples with both methylation and expression data using R tool, *eMap*. The association was considered significant at Bonferroni corrected *p*-value < 0.05. No association was found to be significant for ovarian cancer at this cut-off, and a raw *p*-value < 0.01 was used. R package *ggplot2* was used to generate bubble plots for genes, which were significant in > 30% of the cancers analyzed for each of the kinase family.

### 2.6. Survival Analysis

Survival analysis was carried out using R tools, *survival,* and *survMiner* in the background, for promoter CpGs (±1500 bp from TSS) and the gene expression data. Patients were segregated into high and low expression groups based on the expression median value for each cancer. The β-value cut-off of ≥ 0.6 (high) and ≤ 0.4 (low) was used for the analysis. Cox-regression analysis was performed, and a *p*-value ≤ 0.05 was used as the cut-off to select significant genes and CpG probes. The results were integrated with DE and DM analyses and meQTL analysis to generate the final table. Kaplan-Meier (KM) survival plots were generated by using in-house R code [29]. Logistic regression analysis was performed using R linear model (*lm*) function to categorize tumor and normal samples using the gene expression and methylation data. The classifier performance was measured by calculating the area under the curve (AUC). The receiver operating characteristic (ROC) plots were generated using the *ROCR* R package [30].

## 3. Results

Our analyses were carried out using a set of 496 human kinases that contained DNA methylation, gene expression, and clinical data for all tumor samples analyzed. This kinome set was subjected to unsupervised clustering, differential gene expression and methylation analyses, correlative analysis of gene expression and methylation, and survival analysis, as described below.

### 3.1. Unsupervised Clustering of TCGA and GTEx Samples Based on Kinome Profiles Distinguish Cancer Types

We performed t-distributed stochastic neighbor embedding (t-SNE) method to investigate the pan-cancer gene expression and promoter DNA methylation patterns in kinase genes across 30 TCGA cancers. The t-SNE algorithm is a nonlinear dimensionality reduction technique that is well-suited for embedding high-dimensional data for visualization in a low-dimensional space. The t-SNE clusters represent spatially nearby objects and, therefore, in this case, represent sample similarity. The t-SNE pattern obtained from gene expression and promoter methylation β values of tumor samples (Appendix A), individually, and when combined (Figure 1a–c) showed grouping of cancers without prior knowledge of the sample origin and co-clustering of tumors within an organ system indicating similarity in expression and methylation pattern of kinase genes. This, however, was not true when using a random set of ~500 protein-coding genes (Appendix A), indicating that kinases, despite representing only 2% of the total human genes, are sufficient to distinguish different cancer types. Organ-based solid cancers were observed to be in close proximity; i.e., core gastrointestinal cancers—esophageal carcinoma (ESCA), stomach adenocarcinoma (STAD), colon adenocarcinoma (COAD), rectum adenocarcinoma (READ); cancers of the central nervous system—glioblastoma multiforme (GBM) and brain lower-grade glioma (LGG); thoracic—lung adenocarcinoma (LUAD) and lung squamous cell carcinoma (LUSC); kidney cancers—kidney renal clear cell carcinoma (KIRC), kidney chromophobe (KICH), kidney renal papillary cell carcinoma (KIRP); and this clustering pattern was preserved even after the integration of expression and methylation data types (Figure 1a–c).

A closer, in-depth investigation of the global sample distribution revealed separation of lung, cervical, and esophageal carcinomas into adenoma and squamous type (Figure 1d–f). It is necessary to note that the kinome expression and methylation pattern in esophageal and lung cancers are very similar, as reflected by their co-clustering in the global t-SNE plot.

We wanted to explore further if this was true even for normal sample datasets. We, therefore, analyzed the kinome expression data of 6199 tissue samples from healthy individuals obtained from the Genotype-Tissue Expression (GTEx) project [31], which also revealed easily distinguishable clusters based on the sample origin (Appendix A). Two sub-clusters of the esophagus were observed. One belonging to the esophagus—mucosa, and the other consisted of a mix of esophagus gastroesophageal junction and esophagus muscularis. Two sub-clusters of skin were also observed—one consisting of a mix of sun-exposed (lower leg) and not sun-exposed (suprapubic) and the other from skin cells—transformed fibroblasts. Closer inspection of TCGA datasets also revealed sub-clustering of tumor and adjacent normal samples indicating perturbation of kinase genes in tumor samples with respect to normal tissue. For example, as shown in Appendix A, the normal and tumor tissue form easily distinct subclusters in different cancers showing alteration in the expression and methylation profile during transition from normal to tumor samples.

### 3.2. Differential Gene Expression Analysis

Differential gene expression (DGE) analysis was performed using the R package, *limma* [24]. The analysis was performed for 17 TCGA cancers (BLCA, BRCA, CHOL, COAD, ESCA, HNSC, KIRC, KIRP, KICH, LIHC, LUAD, LUSC, PRAD, READ, STAD, THCA, and UCEC) that have ≥10 adjacent normal samples. Trimmed mean of M-values (TMM) normalization was performed to consider the library size variations among samples [32]. The kinase genes were considered differentially expressed at a false discovery rate (FDR) <0.005 and abs (log_2_FC ≥1) as a cut-off.

The differentially expressed (DE) kinase genes in each cancer are shown in Figure 2. Many kinases, including several dark kinases, were observed to be both commonly upregulated and downregulated across multiple cancers (Figure 2a). Based on the distribution plot of the number of DEG’s obtained, KICH and LUSC had the most downregulated kinases, whereas CHOL had the most upregulated kinases (Figure 2b). The top DE kinase genes found in ≥ 10 (out of 17) cancers are listed in Figure 2c. The most upregulated kinases included Protein Kinase, Membrane Associated Tyrosine/Threonine 1 (PKMYT1) (in 17 cancers), and Maternal Embryonic Leucine Zipper Kinase (MELK) (in 16 cancers). On the other hand, the commonly downregulated included Pyruvate Dehydrogenase Kinase 4 (PDK4) in 15 cancers and KIT proto-oncogene receptor tyrosine kinase (KIT) in 13 cancers, among the cancers analyzed. MELK and PKMYT1 (a dark kinase gene) have been identified as promising therapeutic targets in multiple cancers, such as the brain, colorectal, breast, ovarian, and esophageal cancers, respectively [33,34,35].

Among the 148 dark kinases included in our analysis, PKMYT1 (in 17 cancers), Mitogen-Activated Protein Kinase 15 (MAPK15) in nine cancers, CaM Kinase Like Vesicle Associated (CAMKV) in 8 cancers), PNCK and STK31 (in seven cancers) were commonly upregulated. Similarly, Mitogen-Activated Protein Kinase 4 (MAPK4) in 12 cancers, Serine/Threonine Kinase 32A (STK32A) and P21 (RAC1) Activated Kinase 3 (PAK3) in 10 cancers were found to be commonly downregulated. The complete list of DE genes obtained is provided in Appendix A.

### 3.3. Differential Methylation Analysis

CpG Probes with mean β value difference of at least 0.2 (Δβ ≥ 0.2) at BH adjusted *p*-value < 0.05 between tumor and adjacent normal samples were considered differentially methylated in this study. Differential methylation (DM) analysis was performed for 15 cancers (BLCA, BRCA, CHOL, COAD, ESCA, HNSC, KIRC, KIRP, LIHC, LUAD, LUSC, PAAD, PRAD, THCA, UCEC), which had ≥10 adjacent normal samples with methylation data available.

The distribution of the hyper and hypomethylated probes obtained within 1.5 Kb from the transcription start site (TSS) and the corresponding average gene expression profiles were plotted using the box plot for the analyzed cancers (Figure 3a). A significant difference in the methylation level between the hyper and hypomethylated probes was observed in certain cancers, including BLCA, CHOL, COAD, ESCA, HNSC, KIRC, KIRP, LIHC, LUAD, PRAD, THCA, and UCEC. Lower average expression of the hypermethylated probes in multiple cancers, including significant difference in CHOL, COAD, LUAD, PAAD and UCEC as compared to the average expression levels of the hypomethylated probes, was observed. However, consistent inverse correlation was not reflected in some cancers. An overview of the distribution of differentially methylated (DM) probes is shown in Figure 3b. Overall, more hypermethylated CpG probes were found in PRAD (89%), KIRP (77%), and BRCA (75%), and more hypomethylated probes were observed in LIHC (74.5%), BLCA (72.8%), and THCA (70.2%) cohorts.

Hypermethylation of CpG probes, cg00489401, cg17403609, and cg07682600 mapped to Fms-related tyrosine kinase 4 (FLT4) gene was found in 13, 12, and 11 cancers, respectively. These findings are consistent with studies that identified FLT4 hypermethylation as one of the markers for early/late-stage oral squamous cell carcinoma [36] and cg00489401 (FLT4) as one of the differentiating markers between localized and advanced-stage type 2 Papillary Renal Cell Carcinoma [37].

The CpG probe, cg20994118 (mapped to dark kinase gene, CAMK1G), was commonly hypomethylated in 12 cancers. Previous functional studies of CAMK1G have revealed its role in cell division, mitotic nuclear division, sister chromatid cohesion, cell cycle, and DNA replication. Methylation of the probe, cg20994118, has been reported as negatively correlated with the gene expression on other pan-cancer analysis [38]. The other common DM probes observed in ≥ 10 cancers are listed in Figure 3c. The complete list of DM probes obtained is given in Appendix A.

### 3.4. Correlative Analysis of DNA Methylation and Gene Expression

We used methylation expression Quantitative Trait Loci (meQTL), a correlative analysis, to measure DNA methylation’s influence on gene expression. Pan-cancer analysis of methylation levels of CpG sites within 100 kb of corresponding gene’s TSS was calculated by linear regression model using *eMap1* function in R tool *eMAP V-1.2*. The association was considered significant at Bonferroni corrected *p*-value < 0.05. In the case of ovarian cancer, no association was found to be significant at this cut-off. However, 180 probe-gene pairs were found to be significant at a raw *p*-value < 0.01.

Kinase gene expression can be positively and negatively associated with its corresponding CpG probes since one gene can contain multiple CpGs in the promoter region. As expected, the inverse relationship was predominantly enriched (~92%) within 1.5 kb from the TSS, which can be visualized as a peak in the plot. However, positive correlation was observed to be evenly distributed both up and downstream of TSS (Figure 4a).

An overview of the top 50 dark kinase genes showing a significant correlation among gene expression and DNA methylation across cancers is plotted in Figure 4b. The expression of dark kinase genes was also found to be most negatively correlated with the methylation in the promoter region. Among the most significant correlations ranked by Bonferroni corrected *p*-value, the CpG probes, cg16124934 (RPS6KL1), cg03345668 (MKNK1), cg02133234 (NRBP2), cg04755561 (PKMYT1), cg06532379 (ALPK3), cg27153759 (STK32B), cg13487666 (NEK6) were most frequently occurring. The cancers BRCA, STAD, BLCA, HNSC, and UCEC, showed highly significant negative correlations for most of the dark kinases, as shown in the bubble plot (Figure 4b). The list of significant association within +/−1500 from TSS region from the meQTL analysis is given in Appendix A.

### 3.5. Survival Analysis

Survival analysis was performed using Cox-regression analysis and log-rank test by dividing the patients into high and low expression groups based on the median in each cancer. Several of the commonly upregulated dark kinases (Figure 2) were also significant in survival with a *p*-value < 0.05. Among these, the genes PKMYT1, PNCK, BRSK2, ERN2, and STK31 were significant in survival in five or more cancers (Figure 5b).

Higher expression of PKMYT1 resulted in lower overall survival in ACC, BLCA, KICH, KIRC, KIRP, LGG, and LUAD, and the reverse trend was observed in STAD cohorts. In KIRC patients, survival analysis of corresponding promoter methylation also identified hypermethylation of CpG probe, cg02510853 significant in lower overall survival (*p*-value < 0.001) in KIRC patients (Figure 5a and Figure 6). Association studies using meQTL analysis showed a positive correlation between PKMYT1 gene expression and methylation level of cg02510853 probe (b1 value: 1.374, Adjusted *p*-value: 0.007), which is also reflected in the KM plots. AUC of 0.97 and 0.91 of the PKMYT1 expression and methylation of cg02510853, respectively, suggests its use as a potential diagnostic marker in KIRC patients.

Another commonly upregulated dark kinase gene, whose expression levels were also significant in overall survival is the PNCK gene. Higher expression of PNCK gene resulted in lower overall survival in ACC, GBM, KIRC, KIRP, LIHC, THYM, UCEC, UCS cancer types (Figure 5b). However, no corresponding probes for the same gene were found to be significant for the cut-offs used in our analysis.

We also investigated the role of several commonly downregulated dark kinases in patient survival. One of them included the STK32A gene which was found to be downregulated in 10 cancers. In STAD, the overexpression of STK32A gene (*p* = 0.001) and the hypomethylation of the corresponding CpG probes cg09088988 (*p* = 0.013) was found to be significant in lower overall survival, suggesting its use as a prognostic and diagnostic marker in STAD patients. However, the ROC plots do not indicate its robust use as a diagnostic methylation marker (Appendix A). The expression of STK32A was significantly associated with survival in ACC, CESC, LUAD, and PCPG cohorts.

Similarly, overexpression of MAPK4 and hypermethylation of cg19448837 was found to be significant (*p* < 0.005) in lower overall survival in STAD patients (Appendix A). We also found high and low expression groups of MAPK4 have a significant difference in the overall survival with a *p*-value < 0.05 in LUAD, LUSC, PAAD, THCA, UCEC, and UVM patients. Multiple MAPK4 CpG probes, namely cg05492442 and cg20068620, were significant in LGG patients (Figure 5a).

Both expression and methylation, high vs. low groups, were significant for several dark kinases, notably Obscurin, Cytoskeletal Calmodulin And Titin-Interacting RhoGEF (OBSCN), Serine/Threonine Kinase 3 (STK3) and MAPK4 kinases, which are displayed in Figure 5a. We found five CpG probes mapped to OBSCN gene significant in survival in KIRP, UCEC and UVM cancers. OBSCN gene has been reported to interact with many cancer-associated genes involved in breast tumorigenesis [39]. The survival analysis results based on expression and methylation are given in Appendix A respectively.

## 4. Discussion

Classification of tumors based on high throughput data has been achieved using datasets such as gene expression, CpG island methylator phenotype (CIMP) status alone, or by integrating it with other high-throughput data types using various approaches [40,41,42]. A similar grouping of cancers has also been achieved using transcription factors’ gene expression data [43]. Few computational studies have reported tissue-specific expression of kinases that are not significantly enriched in any central or peripheral tissue types [4]. Therefore, we asked if a similar grouping of cancer types can be achieved by an unsupervised method using only the differences in the transcriptional and methylation activity of the kinome gene set. However, unlike most of these studies that use only one type of omics data or use the genome-wide gene set, in this study, we demonstrated that organ-system, tumor, and subtype level classification could be achieved and preserved after the integration of expression and corresponding promoter methylation data for nearly 8000 tumor samples across 32 cancer types (Figure 1a–c). The same clustering pattern was observed in normal tissues (GTEx), which suggests that the kinome expression and methylation profiles are unique to the tissue type irrespective of the disease condition. As a control dataset, we wanted to evaluate if the observed grouping of samples based on organ system, cancer types, or histological subtypes is not an intrinsic property of tumors. Replicating the unsupervised clustering analysis using 500 randomly selected protein-coding genes did not yield similar results, indicating that the observed grouping can only be attributed to the expression and methylation pattern of kinase genes (Appendix A).

A closer look at the t-SNE plots also showed distinct histological tumor subtypes such as adenocarcinoma and squamous cell neoplasms of lung, esophagus, and cervical cancer (Figure 1d,e). This is consistent with the results obtained using genome-wide data by Lin et al. [44], indicating histology-driven differences in expression, methylation, and pathways and upstream regulators may be consistent across anatomical boundaries and are also true for the kinome gene set.

To further identify the commonality and differences in kinase genes’ expression and methylation pattern in various cancers, we performed DGE and DM analysis of kinase genes and corresponding probes in the promoter region. We further highlighted the results of relatively poorly understood or dark kinase genes to provide potential starting points that can be explored as a potential target. To identify the clinical relevance, we also performed meQTL and survival analyses followed by classification using linear regression models to identify prognostic and diagnostic relevance.

The DGE analysis showed the genes PKMYT1, and MELK upregulated in 17 and 16 cancers, respectively (Figure 2). MELK is a member of AMPK/Snf1 family of serine/threonine kinases, and the MELK protein expression is highly specific to proliferating cancer stem cells [45]. Studies have shown that high expression of this serine/threonine kinase is associated with poor patient prognosis. MELK disruption is found to inhibit tumor growth and trigger cell cycle arrest in breast cancer cells [46]. Given the preferential upregulation of MELK in various cancer types, small molecule inhibitors of MELK have been developed and are currently in Phase I clinical trials for metastatic breast cancer [47]. From our expression-based survival analysis results, MELK was found to have a significant role in survival in 10 different cancers, including ACC, KICH, KIRC, KIRP, LGG, LIHC, MESO, PAAD, THYM, and UVM. MELK inhibitor, OTS167, is reported to suppress tumor growth in breast, lung, prostate, and pancreatic cancer cell lines [48]. However, due to its extremely unselective nature, studies have contradicted its use for the clinical validation of MELK, suggesting that its biological activity may not be attributed to MELK inhibition alone [9].

PKMYT1, a dark kinase, is a member of Wee family of tyrosine/threonine kinases and shares high functional similarity with WEE1 [49]. PKMTY1 is a key regulator of the cell cycle complex and plays an important role in tumor progression [50,51]. The CpG probe, cg04755561 (mapped to PKMYT1), was found to be significantly negatively correlated with the gene expression from the meQTL analysis in 13 cancers. However, we did not find its role in survival in our study. Survival analysis also showed overexpression of PKMYT1 leads to lower overall survival in ACC, BLCA, KICH, KIRC, KIRP, LGG, and LUAD cancer types (Figure 5b). Other computational studies have also identified PKMYT1 as a prognostic marker in kidney cancer cohorts [11]. Besides, we identified that hypermethylation of the CpG probe, cg02510853 leads to lower overall survival in KIRC patients (Figure 5a and Figure 6). Interestingly, PKYMT1 is a potentially druggable kinase that has been implicated in the survival of GBM-like stem cells, indicating that it could be explored as an actionable target [51,52] in other cancers.

Other less understudied commonly upregulated kinases that were frequently found to be significant in survival (≥5 cancers) included the BRSK2, ERN2, PNCK, and STK31 kinases (Figure 5b). BRSK2, a member of the AMPK-related family of kinases, has been shown to negatively regulate nuclear factor erythroid 2-related factor (NRF2) based on the gain of function kinome screen [53]. It has also been shown to positively correlate with PDAC metastasis, promoting neoplastic cells’ invasiveness in nutrient-deprived conditions [54]. BRSK2 overexpression can be linked to poor survival in LIHC, KIRC, KIRP, LGG, THCA and UCEC cancer types (Figure 5b) from our survival analysis. Therefore, BRSK2 is a promising prognostic marker that can be further explored. PNCK knockdown has been reported to regulate PI3K/AKT/mTOR signaling pathway and suppress growth and induce apoptosis of nasopharyngeal carcinoma cells in vitro and in vivo [55]. Co-overexpression of HER-2 and PNCK has also been known to enhanced tumor cell proliferation and Trastuzumab resistance [56].

ERN2, a serine/threonine kinase, was overexpressed in KIRP, STAD, THCA, and UCEC and downregulated in PRAD. We identified overexpression of ERN2 and other kinases including NIMA-related kinase 2 (NEK2), serine/threonine/tyrosine kinase 1 (STYK1), and polo-like kinase 1 (PLK1) in a subset of 146 pancreatic ductal adenocarcinoma (PDAC) patients after removing endocrine, invasive adenocarcinoma, undifferentiated, or mixed pancreatic cancers from TCGA-PAAD samples [57]. Overexpression of ERN2 has earlier been reported 60–70% of colorectal samples [58]. Recently, ERN2 was found differentially expressed in different mediastinal lymph node metastasis (MLNM) in lung adenocarcinoma [59]. However, our analysis suggests that ERN2 can be explored as a potential target in multiple cancer types despite limited literature support.

The DM analysis shows significant differences in the kinase methylation levels between tumors and adjacent normal samples in BLCA, CHOL, COAD, ESCA, HNSC, KIRC, KIRP, and LIHC LUAD, PRAD, THCA, and UCEC were observed. However, the expected inverse methylation-expression relationship was not reflected in some cancers when we plotted the corresponding gene expression values (Figure 3a). This is probably because of other genetic and epigenetic factors that regulate the expression pattern and might not be a direct consequence of changes in the DNA methylation status alone. However, further in-depth correlation analysis showed that the inverse relationship was found to be more enriched within 1.5 kb from the TSS. The peak indicates a more significant correlation with a lower *p*-value in the plot (Figure 4a). Finally, this pan-cancer analysis uncovered the potential value of many novel kinases, including several dark kinases as novel prognostic and diagnostic markers owing to their association with survival. The findings of this study could be coupled with the sequencing of a patient’s genome to enhance cancer detection, tumor prognosis, and prediction to treatment and response. Hence, this study paves the way for further investigation and experimental validation of novel kinase targets for their potential therapeutic use in multiple cancers.

## 5. Conclusions

Many kinases are associated with cancer initiation and progression. However, only a small proportion are currently being targeted due to a lack of characterization of their biochemical and biological functions. Therefore, it is important to prioritize and identify newer kinase targets. In this study, we showed kinase-based clustering of the tumor and normal samples based on their organ system and tissue histology, which revealed commonality and uniqueness of expression and methylation profiles among different cancer types. We also demonstrate that the gene expression and DNA methylation profiles of the kinome alone, independently, or combined are sufficient to achieve the above grouping. While this pan-cancer study reiterated the importance of known kinase targets, we also demonstrated that several novel dark kinases due to their strong association to survival could serve as prognostic and diagnostic biomarkers across multiple cancers. In conclusion, our study paved the way for the therapeutic characterization of many poorly understood dark kinase genes (PKMYT1, PNCK, BRSK2, ERN2, STK31, STK32A, MAPK4) that need to be investigated further.

## Figures and Tables

**Figure 1 cancers-13-01189-f001:**
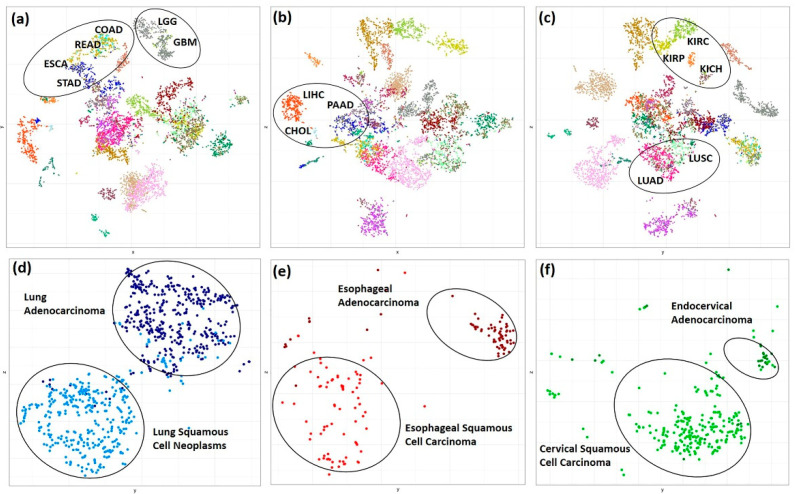
Distribution of the TCGA cancer samples in 2D t-SNE plot. (**a**) x vs. y (**b**) x vs. z (**c**) y vs. z coordinates for 6270 (combined kinase gene expression and promoter CpGs β values) features and 7783 TCGA tumor samples belonging to 30 cancer types. The mapped data points are colored as per the cancer types. (**d**–**f**) 2D t-SNE plot showing separation of TCGA Esophageal, Lung Carcinoma and Cervical Cancer into Adeno and Squamous histological tissue types based on combined kinase gene expression and methylation data of 930 samples which includes 71 esophageal adenocarcinoma, 80 esophageal squamous cell carcinoma, 414 lung adenocarcinomas, 365 lung squamous cell carcinoma, 246 cervical squamous cell carcinoma and 30 endocervical adenocarcinoma.

**Figure 2 cancers-13-01189-f002:**
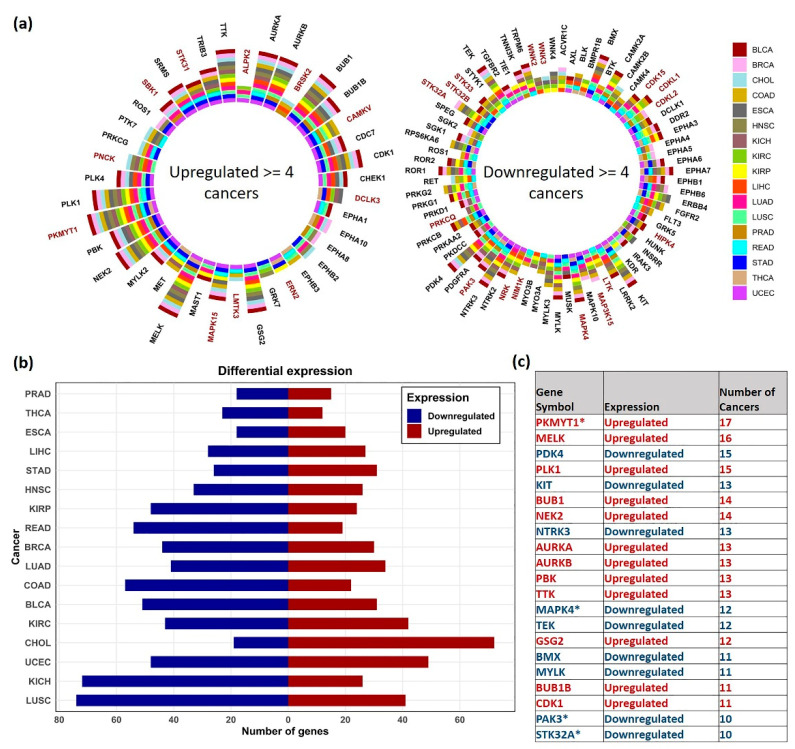
Distribution of the differentially expressed genes. (**a**) Sunburst plot showing the top DE kinase genes in ≥ 4 TCGA cancer types. Dark kinases are highlighted in maroon (**b**) Number of Upregulated (red) and downregulated (blue) kinase genes observed in analyzed cancers. (**c**) Table showing the list of common DE genes obtained in ≥ 10 cancers, the direction (upregulated—red, downregulated—blue) and in number of cancers observed. Dark kinases are marked with * symbol.

**Figure 3 cancers-13-01189-f003:**
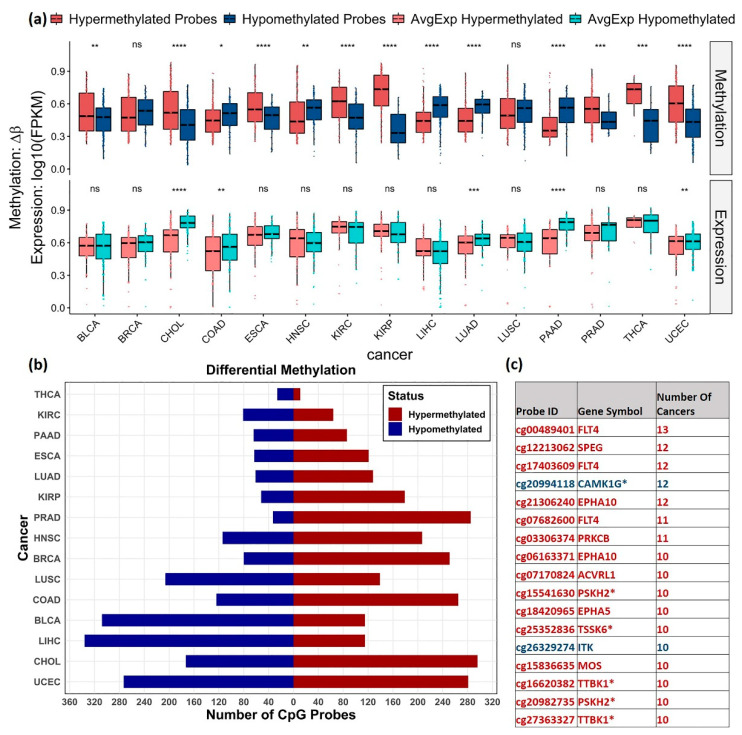
Distribution of DM CpGs across cancers. (**a**) Box plots showing distribution of hyper (Red) and hypomethylated (blue) probes and the corresponding average gene expression (light red and light blue) in different cancers. The gene expression values were normalized between 0 and 1. T-test was used to show the significance level between the methylation levels of the hyper and hypo methylated probes and between the corresponding gene expression level (ns: *p* > 0.05, *: *p* ≤ 0.05, **: *p* ≤ 0.01, ***: *p* ≤ 0.001, ****: *p* ≤ 0.0001). CpG probes with mean β value difference of at least 0.2 (Δβ ≥ 0.2) at BH adjusted *p*-value < 0.05 were considered differentially methylated. (**b**) Distribution of hypermethylated (red) and hypomethylated (blue) probes obtained for each cancer. (**c**) List of commonly observed probes DM in ≥ 10 cancers, their direction of methylation (hypermethylation—red and hypomethylation—blue) and the number of cancers observed. Dark kinase genes are marked with * symbol.

**Figure 4 cancers-13-01189-f004:**
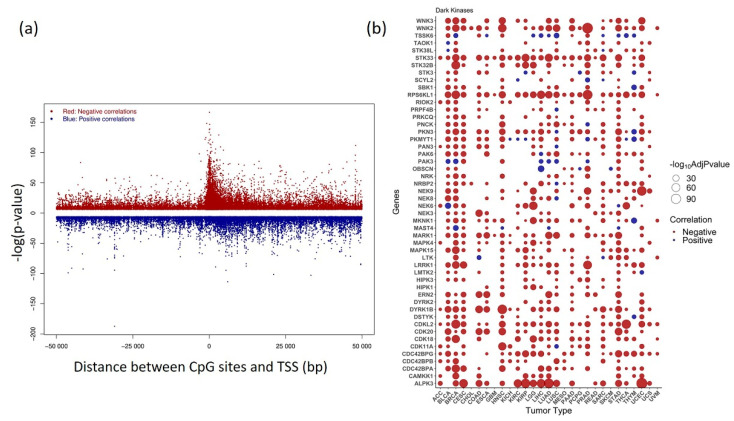
Correlation between kinase gene expression and methylation (**a**) Distribution of pan-cancer positive and negative correlation between DNA methylation β value and gene expression plotted against distance between CpG sites within 50Kb and transcription start site (TSS) (**b**) Bubble plot showing the status of most significant correlations belonging to Dark Kinases group +/− 1500 bp from TSS obtained at Bonferroni corrected *p*-value < 0.05. Negative correlations are shown in red and positive correlations are shown in blue. Genes with significant correlation values obtained in > 30% of analyzed cancers are plotted.

**Figure 5 cancers-13-01189-f005:**
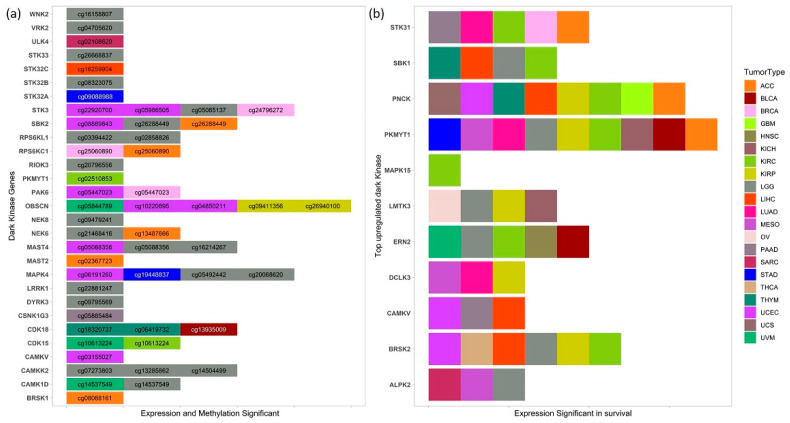
Dark kinase genes and methylation in survival (**a**) List of dark kinases whose expression, and methylation of CpG probes found to be significant in survival (*p*-value < 0.05) in various cancers. (**b**) Top upregulated dark kinases from DGE analysis whose high and low expression groups also have a significant difference in the overall survival (*p*-value < 0.05) in several cancers.

**Figure 6 cancers-13-01189-f006:**
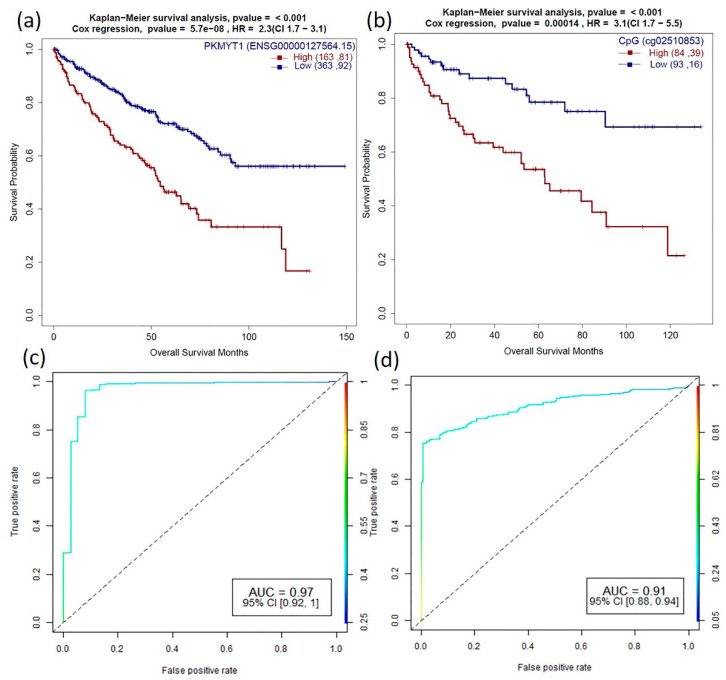
Role of PKMYT1 in prognosis and diagnosis (**a**,**b**) Survival plots of PKMYT1 high vs. low gene expression and promoter DNA methylation sites (cg02510853) which are associated with KIRC patient survival with *p*-value for KM plot (log-rank test) and Cox proportional hazard model. (**c**,**d**) Corresponding ROC plot of gene expression and promoter methylation for the generalized linear model.

## Data Availability

Publicly available datasets were analyzed in this study. The TCGA level-3 data is available at https://portal.gdc.cancer.gov/projects/ (accessed on 1 March 2020). GTEx data are available at https://www.gtexportal.org/home/datasets (accessed on 1 March 2020).

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
