# Peer review of "Pan-Cancer Analysis of Human Kinome Gene Expression and Promoter DNA Methylation Identifies Dark Kinase Biomarkers in Multiple Cancers"

_cancers, 2021, doi:10.3390/cancers13061189_

Round 1
Reviewer 1 Report
Southekal, et al. present here a study where they use gene expression data and DNA (promotor) methylation patterns within the human kinome to identify kinase biomarkers in various cancers. The study identifies prognostic/diagnostic biomarkers and focuses on uncharacterized (or dark) kinases to encourage research on such kinases as therapeutic targets. The manuscript is well-written and provides a convincing comprehensive analysis of human kinome profiles, as well as the extent of methylation in promotor regions of up- and down-regulated kinases, in several cancerous tissues. This study will be of general interest to the readers of Cancers and it is this reviewer’s opinion that publication is warranted. However, there are a few minor revisions outlined below that need to be addressed in order to enhance the overall impact of the study:
1) The authors state that many different kinases - including several dark kinases - were observed to be both up- and down-regulated across multiple cancer types. However, they do not explicitly mention of what types of kinase regulation (and/or DNA methylation) patterns are observed in the normal tissues of specific patients. To be sure, the GTEx data only shows clustering of kinases for whole populations. Contemporary views on cancer treatments are veering towards applying more precision medicine and it seems that the authors’ current approach to identifying cancer biomarkers could serve to complement personalized therapies. While such a comprehensive study is obviously outside the scope of this paper, it would significantly enhance the impact of the current study if the authors briefly discussed how their approach could be coupled with the sequencing of a patient’s genome to enhance precision treatments.
2) The data mining approach used by the authors has led to the identification of kinases (even some dark kinases) that are up- and down-regulated in specific cancers. While this information is extremely useful in identifying novel targets for cancer treatments, it does not speak to the difficulty of targeting discrete kinases with chemotherapies. To be sure, this information will only facilitate treatments once we learn how to target kinases without broad-spectrum inhibition. The authors should discuss more fully how their data may allow for the development of novel therapies that can target specific kinases with higher precision.
3) The authors state that their comparative GTEx analysis shows that kinome and methylation profiles are unique to the tissue type irrespective of the disease condition. This leads to an important question regarding the dark kinases: Can the authors’ data be used to identify the function of specific dark kinases? If so, the authors could speculate on this or, at the very least, mention how the current data set could be used to dissect signaling pathways that are regulated by dark kinases of unknown function in specific tissues.
4) A comprehensive Table that lists the up- and down-regulated kinases and promotor methylation patterns in different cancers would be a helpful addition to the main text. This would serve to rapidly communicate their primary results to the reader.
5) The axis label text in Figs 3 and 4 are unreadable and need to be in a larger font size.
Author Response
"Please see the attachment."

Reviewer 2 Report
The manuscript of Southekal examined human kinase (kinome) gene expression and compared this to DNA methylation patterns obtained from TCGA cancer data. The authors used unsupervised clustering to determine how the expression patterns tracked with different cancer types and patient outcome. Both hyper and hypo methylation patterns were examined from tumor and adjacent tissue. Methylation expression quantitative trait loci (meQTL) analysis revealed correlative and anti-correlative associations between transcription start site (TSS) methylation patterns and kinase expression. Importantly the authors analyses pinpointed a number of understudied or “dark” kinases and connected their expression data to select cancers for the first time. Their analyses suggest that kinases such PKMYT1, BRSK2, ERN2, MAPK4 and others may have a significant role as a diagnostic or potential target for certain types of cancer. Overall this is an important study that provides significant new insight into kinase expression, particularly the understudied kinases, and their potential role in disease.
Critique
This is a well- written manuscript- only minor changes are necessary.
- While possibly outside the scope of this study it would be interesting if the authors would comment on how their data overlays with kinase mutation data, in particular with the dark kinases.
- In the discussion (p.12), the authors state that OTS167 could be used to validate MELK as a therapeutic target for cancer.OTS167 has been shown to be a highly non-selective inhibitor (Klaeger et al., 2017) and inhibits many kinases more potently than MELK. Hence studies have refuted its use as proof of MELKs role in cancer. This sentence should be re-written to more accurately reflect the information on MELK.
Author Response
"Please see the attachment."

Reviewer 3 Report
The present study explores the global human kinome expression and corresponding promoter DNA methylation profiles across 32 TCGA cancer types to study the altered kinome expression profiles and correlate with corresponding promoter methylation status. The authors have identified potential kinase biomarkers with clinical relevance, in particular commonly upregulated dark kinases, PKMYT1, Pregnancy Up-Regulated Nonubiquitous CaM Kinase (PNCK), BRSK2, Endoplasmic Reticulum To Nucleus Signaling 2 (ERN2), and Serine/Threonine Kinase 31 (STK31. The study is well written and well structured. The authors performed a very thorough and complex analysis on the public TCGA data. This element is certainly a way to make the TCGA projects usable and differently interpretable. The figures and tables aid in the understanding of the study.
Author Response
"Please see the attachment."
